# On the n-Dimensional Phase Portraits

**Martín-Antonio Rodríguez-Licea [1,*,†]** , **Francisco-J. Perez-Pinal [2,†]** , **José-Cruz Nuñez-Pérez [3]** and **Yuma Sandoval-Ibarra [3]**

[1]  Departamento de Ingeniería Electrónica, CONACYT-Instituto Tecnológico de Celaya, Guanajuato 38010, Mexico
[2]  Departamento de Ingeniería Electrónica, Instituto Tecnológico de Celaya, Guanajuato 38010, Mexico; francisco.perez@itcelaya.edu.mx
[3]  Instituto Politécnico Nacional, CITEDI, Tijuana BC 22435, Mexico; nunez@citedi.mx (J.-C.N.-P.); jumasaniba@gmail.com (Y.S.-I.)
*   Correspondence: marodriguezl@conacyt.mx
†   These authors contributed equally to this work.

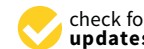

**Featured Application: The current graphic analysis can be applied during the mathematical analysis of high order systems, for instance, power electronic, mechanical, aeronautic, and nuclear plant systems among others.**

**Abstract:** The phase portrait for dynamic systems is a tool used to graphically determine the instantaneous behavior of its trajectories for a set of initial conditions. Classic phase portraits are limited to two dimensions and occasionally snapshots of 3D phase portraits are presented; unfortunately, a single point of view of a third or higher order system usually implies information losses. To solve that limitation, some authors used an additional degree of freedom to represent phase portraits in three dimensions, for example color graphics. Other authors perform states combinations, empirically, to represent higher dimensions, but the question remains whether it is possible to extend the two-dimensional phase portraits to higher order and their mathematical basis. In this paper, it is reported that the combinations of states to generate a set of phase portraits is enough to determine without loss of information the complete behavior of the immediate system dynamics for a set of initial conditions in an n-dimensional state space. Further, new graphical tools are provided capable to represent methodically the phase portrait for higher order systems.

**Keywords:** high order system; n-dimensional; phase portrait

## 1. Introduction

Historically, humanity has tried to understand, explain and represent its surrounding environment by using several tools familiar with its perception of reality. Numerous examples of exceptional theories and enlightenment of natural incidents have been explained by instruments provided by different civilizations as the ancient Egyptian, Mesopotamian, Greek, Chinese, Maya, etc. [1]. Up to date, those tools in combination with theories and information accumulated through the years has allowed us to represent and analyze complex physical events in a manageable manner by using, among others, mathematical models. A model, in general terms, is a mathematics representation to study a phenomenon in the best conditions of space, time and cost [2]. Indeed, modeling complexity increases in terms of analysis information needs. For instance, there are several levels of modeling as macroscopic, microscopic (non-spatial, spatial), and submicroscopic. The interested reader is referred for more details about modeling to reference [3].

Particularly, in control theory, two basic and widely used visual analysis tools are based on Poincaré maps and phase portraits. A discrete time system, derived from the original continuous system with the main aim to reduce its complexity is referred as Poincaré map [4]. On the other hand, a pictographic tool used to represent the instantaneous state of two interest variables in any autonomous system is referred to as phase portrait; this paper deals with the second kind of representation.

A regular phase portrait is a two-dimensional, qualitative representation of the system dynamics with an implicit order reduction, information related to time is lost and only the instantaneous behavior is presented for a set of initial conditions [5]. Such initial conditions are represented unambiguously as points in an $\mathbb{R}^2$ plane. From each point, an arrow directed towards the corresponding derivative with size proportional to its magnitude is drawn (e.g., Figure 1).

This representation can be extended to $\mathbb{R}^3$ phase portraits; however, some information about the state variables is lost. The three-dimensional phase portrait is turned two-dimensional by a snapshot in order to be printed out and one must select the best view of the 3D-to-2D converted phase portrait or look for an additional degree-of-freedom for the plot, as color or time (by generating a video). Some authors have preferred to graph the solution of the system for a representative initial condition instead of making a phase portrait e.g., Figure 3. For nonlinear systems, an unexpected dynamic behavior can be lost sight since the point of view of the snapshot or the selection of the initial condition rely on the appreciations capacity.

Some authors have also intuitively presented multiple phase-portraits for 3 and 4 state variables, by pair-combining [6–9]; however, there is still the question of whether it is possible to extend this idea to higher order systems and its mathematical basis. A brief historical background of phase portrait for dimensions greater than two is given as follows.

A first effort to represent phase plane of higher order was analyzed in [10]. In that study, was proposed the use of a $x^n$ vs. $x$ plane analysis, where the Poincaré's fundamental phase plane is contemplated as the simplest case of the general proposed method. A $\mathbb{R}^3$ dimensional case was analyzed in [11]. The authors in that study used as an example, a harmonic oscillator, and also used a Wigner Transformation as an order reduction tool to simplify the system complexity. Another interesting approach was proposed in [12]. In that study, an oriented linear graph for piecewise-linear system called "generalized phase portrait" was proposed. A three-phase system was analyzed, but examples for a larger dimension were not reported. A computational program to represent an n-dimensional system in phase-space was analyzed in [13]. The reported system consisted on a phase space navigator and a user interface, and it allowed to accomplish the following tasks: simulate the proposed system with different initial conditions, plot trajectories in a paper sheet, and to design controller laws. Unfortunately, that paper reported a maximum of a three-dimension system. Another interesting computational tool based on SIMULINK was analyzed in [14]. The study regarded piecewise-linear systems and represented them as a set of convex polytopes. A tetrahedron was used to represent a four-dimension system, and the "hyperplane" term was defined. Limiting the capacity of the proposed system up to five states, as stated by authors, was the main limitation. On the other hand, a three-dimensional case for phase portrait and some explicit formulas by using Lyapunov exponents was given in [15]. Indeed, the main contribution of that study was to calculate analytically a Lyapunov exponent from three dimensional quadratic mappings. Nevertheless, extension of this methodology for more than this case was not stated. An hyperchaotic four-wing dimensional system was analyzed in [16]. Phase portrait for a combination of the four states, numerical and practical results were reported. However, applications of the proposed system were not documented.

As can be noticed from the previous discussion, there is still the question of whether it is possible to extend phase portraits to a higher order and the mathematical basis from the reader's perspective. In summary, these studies highlight the need for developing a standardized graphical tool capable to represent, methodically in a single stage, the overall dynamics of complex systems. Indeed, a systematic approach for representing more than $\mathbb{R}^2$, to the author's knowledge, has not yet been reported. Therefore, in this paper is proposed a method to address this question. Initially, it is shown that using

combinations of states to generate a set of phase portraits is enough to determine without loss of information the complete behavior of the immediate system dynamics for a set of initial conditions. Further, are provided three graphical tools capable to represent methodically the phase portrait for higher order systems including alternatives that aim to maintain only relevant information on the dynamics and simplifying the phase portrait construction.

This paper is organized as follows. In Section 2 are reported foundation, procedure, and some examples to plot the phase portraits by state combinations. In Section 3, the same topics to plot the phase portraits by a coordinates transformation (order reduction) are reported. In Section 4, an n-dimensional phase portrait by a state-by-state plot is reported. In Section 5, final conclusions are given.

## 2. The n-Dimensional Phase Portraits by State Combinations

In this paper is assumed that the reader has the basic notions of non-linear second order systems and the initial condition problem. For a very comprehensible explanation of two-dimensional phase portraits and its construction the reader is referred to [5].

A regular phase portrait is a two-dimensional representation of the system dynamics with an implicit order reduction: information with respect to the time is lost and only the instantaneous behavior is presented for a set of initial conditions. Such initial conditions are represented unambiguously as points in an $\mathbb{R}^2$ plane, and a set of arrows directed toward derivative and with size proportional to its magnitude are originated in such points.

As mentioned before, some authors have informally presented multiple phase-portraits by pair-combining for 3 and 4 variables. However, one can question if it is enough to study a system completely and without loss of information with such approach, for higher dimensions; that is, one must ensure that looking an n-dimensional phase portrait from every 2D possible perspective is enough.

### 2.1. Foundation

This section aims to obtain and model the overall possible perspectives that an observer/onlooker can have of a multidimensional object.

In particular, a point $x \neq 0$ in a 2-dimensional Euclidean plane (with axis $X_1$ and $X_2$), is viewed by an onlooker located at a distance $r$, with a perpendicular sight to the plane in that point, univocally; that is, every two-dimensional characteristic is visible and measurable regardless of the sharpness. This approach is extended for a sufficiently small line and its centroid for n-dimensional and the pair plane-onlooker is named view.

**Definition 1.** *An n-point or a small enough n-line (n-sline) in a $n > 2$ dimension can be determined univocally, if every n-dimensional characteristic is uniquely measurable by an onlooker.*

**Definition 2.** *A non-empty set of n-points and n-slines in conjunction with some Euclidean plane is a view.*

**Definition 3.** *The sharpness of a view is the capacity of the onlooker to differentiate two or more n-points and/or n-slines in a view.*

Clearly, most of humans can imagine and determine an object univocally for dimensions less or equal to 3. For higher order dimensions this is not a trivial task; for instance, to determine univocally an object in a dimension 6: although mathematics can easily handle such higher dimensions, it is unnatural for a human being to analyze such object.

In the following, it is demonstrated that an onlooker for each plane (e.g., $X_1 - X_2$, $X_1 - X_3$, and $X_2 - X_3$ for $\mathbb{R}^3$) is enough to determine univocally the 3-dimensional characteristics of a set of n-points and n-slines, specifically:

**Theorem 1.** *An n-point needs $\nu = \frac{n!}{2(n-2)!}$ views to be determined univocally.*

**Proof.** The number of different 2D planes of an n-dimensional phase space is equal to the number of combinations to select different subsets $(x_i, x_k)$ out of the set $\{x1, x2, ....xn\}$, which is the binomial coefficient $\mathcal{C}(\cdot, \cdot)$ of pairs of coordinate-axis. That is:

$$\nu = \mathcal{C}(n, 2) = \frac{n!}{2(n-2)!} \tag{1}$$

☐

For a nonempty set of n-points or n-slines, one can easily extend the previous result.

**Lemma 1.** *A nonempty set of n-points or n-slines needs $\nu = \frac{n!}{2(n-2)!}$ views with enough sharpness to be determined univocally.*

**Proof.** From the fact the onlooker can distinguish different n-points and n-slines (enough sharpness) in a view and Theorem 1, then the conclusion is obtained.  ☐

*2.2. Procedure*

From the above, a first approach to construct phase portraits for higher order systems and be determined univocally, is by analyzing separately each combination of states as described in the following.

**Procedure 1.** *Follow the next steps:*

1. *Construct the state space representation of the system.*
2. *Define regular initial conditions' values for each state ().*
3. *Construct a phase portrait for each combination of states, a total of $\left( \frac{n!}{2(n-2)!} \right)$.*
4. *If necessary, redefine the initial conditions' values and repeat step 3; that is, get enough sharpness.*
5. *Analyze each phase portrait separately.*

The above procedure is a generalization of the 2D phase portrait as presented in [5]. For $n = 2$, the Step 2 reduces to construct a grid of initial conditions; this is, a vector of 2 columns and $y$ rows of initial conditions where $y$ depends on the grid size and spacing. For higher $n$ dimensions, a vector with $n$ columns must be proposed and the quantity of rows depends again on the n-grid size and spacing. For the Step 3, unlike a 2D portrait, one will get a set of arrows that arise from each initial condition instead of just one. It may be convenient to use a different color for each initial condition in order to differentiate the arrows that arise from each initial condition. In the following, some examples are provided.

*2.3. Examples*

Consider the stable system

$$\dot{x}_1 = -x_1$$
$$\dot{x}_2 = -x_2$$
$$\dot{x}_3 = -x_3 \tag{2}$$

Accordingly to the Theorem 1, three views are enough to determine univocally the 3-dimensional characteristics. Following the Procedure 1, this is already a state space representation such that the first step is done. Next, for the Step 2 a n-grid is proposed with a stepping of 10 and from $-30$ to 30 units:

| $x_1(0)$ | $x_2(0)$ | $x_3(0)$ |
|---|---|---|
| −30 | −30 | −30 |
| −30 | −30 | −20 |
| −30 | −20 | −30 |
| −30 | −20 | −20 |
| ⋮ | ⋮ | ⋮ |
| 30 | 30 | 20 |
| 30 | 30 | 30 |

For the Step 3, $\dot{x}_1$, $\dot{x}_2$, and $\dot{x}_3$, as well as the length of the arrows ($len(a,b) = \sqrt{a^2 + b^2}$ function) are calculated for every row:

| $x_1(0)$ | $x_2(0)$ | $x_3(0)$ | $\dot{x}_1$ | $\dot{x}_2$ | $\dot{x}_3$ | $len(\dot{x}_1, \dot{x}_2)$ | $len(\dot{x}_1, \dot{x}_3)$ | $len(\dot{x}_2, \dot{x}_3)$ |
|---|---|---|---|---|---|---|---|---|
| −30 | −30 | −30 | 30 | 30 | 30 | 42.4 | 42.4 | 42.4 |
| −30 | −30 | −20 | 30 | 30 | 20 | 42.4 | 36 | 36 |
| −30 | −20 | −30 | 30 | 20 | 30 | 36 | 42.4 | 42.4 |
| −30 | −20 | −20 | 30 | 20 | 20 | 36 | 36 | 28.3 |
| ⋮ | ⋮ | ⋮ | ⋮ | ⋮ | ⋮ | ⋮ | ⋮ | ⋮ |
| 30 | 30 | 20 | −30 | −30 | −20 | 42.4 | 36 | 36 |
| 30 | 30 | 30 | −30 | −30 | −30 | 42.4 | 42.4 | 42.4 |

The first view ($x_1 - x_2$) is obtained by plotting a classic phase portrait with the columns $x_1(0)$, $x_2(0)$ as the points of the grid, $\dot{x}_1$, $\dot{x}_2$ as the destination point (direction) of the arrows, and $len(\dot{x}_1, \dot{x}_2)$ as the length of the arrow. The second and third views are obtained similarly, and the full phase portrait consists of the 3 views shown in Figure 1. Note that this phase portrait allows to see with no doubt that the system can be asymptotically stable for the set of initial conditions by inspecting each view separately; that is, all of the views should look like a phase portrait with an apparently stable equilibrium point. Recall that many math software include automated 2D phase portrait functions (for instance "'quiver'" in Matlab, see Appendix A for a basic code).

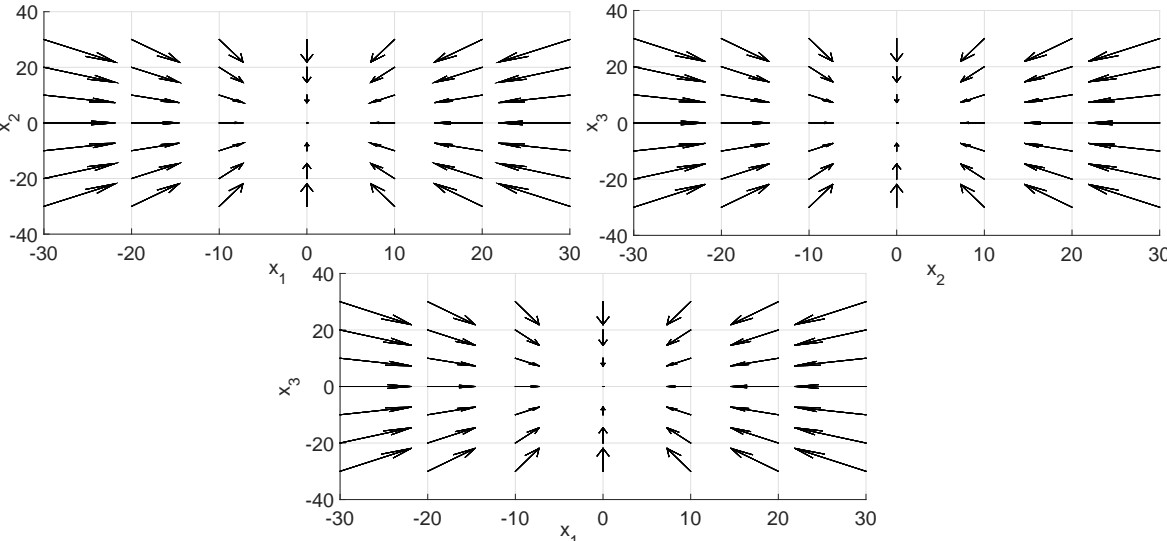

**Figure 1.** Phase portrait views for the system (2).

Now consider the chaotic system [17]

$$\dot{x}_1 = 35(x_2 - x_1) + x_2 x_3$$
$$\dot{x}_2 = 25x_1 - x_2 - x_1 x_3$$
$$\dot{x}_3 = x_1 x_2 - 7x_3 \tag{3}$$

For such system, the phase portrait views are shown in Figure 2 and in this case the trajectories of the system are plotted for some initial conditions with colors. The author in [17] plotted the systems' trajectories in a 3D phase plane (reproduced as Figure 3 in this paper) for only a single initial condition. Note that the n-dimensional phase portrait presented in this study allows a clearer phase portrait representation of a set of initial conditions and trajectories for a subset of initial conditions.

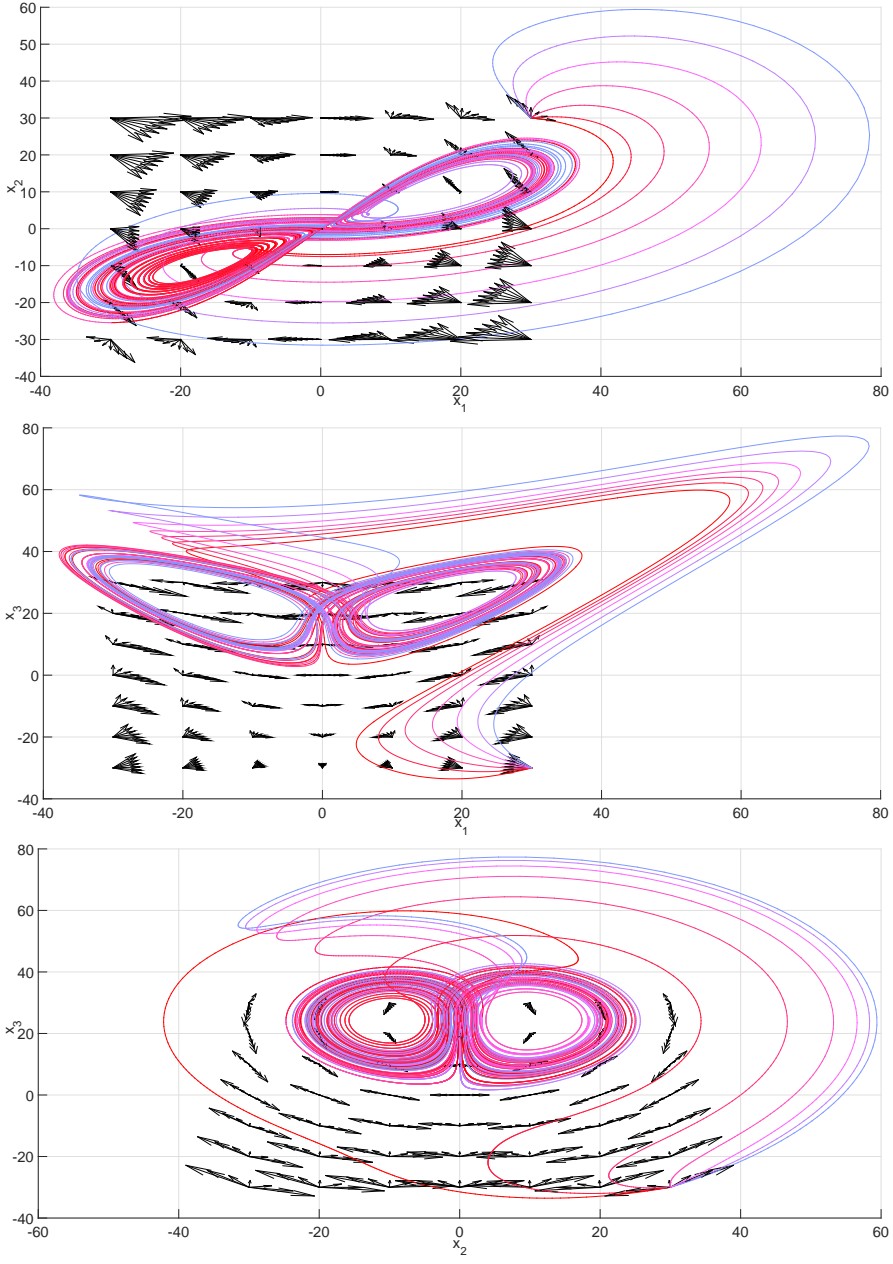

**Figure 2.** Phase portrait views for system (3).

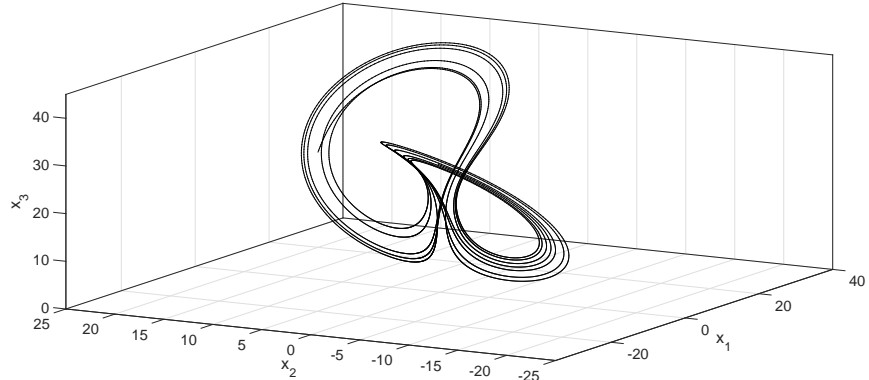

**Figure 3.** 3D Phase solution for $x_1(0) = x_2(0) = x_3(0) = 20$ for system (3) [17].

Now consider the arbitrary system

$$\dot{x}_1 = Sin(x_1)$$
$$\dot{x}_2 = Cos(x_2)$$
$$\dot{x}_3 = 0.1 * Tan(x_3)$$
$$\dot{x}_4 = Abs(x_4) \tag{4}$$

In Figures 4–9 are shown the views for such system. Clearly one can perceive several stable and unstable points. In Appendix A is shown a Matlab code example to generate the six views automatically.

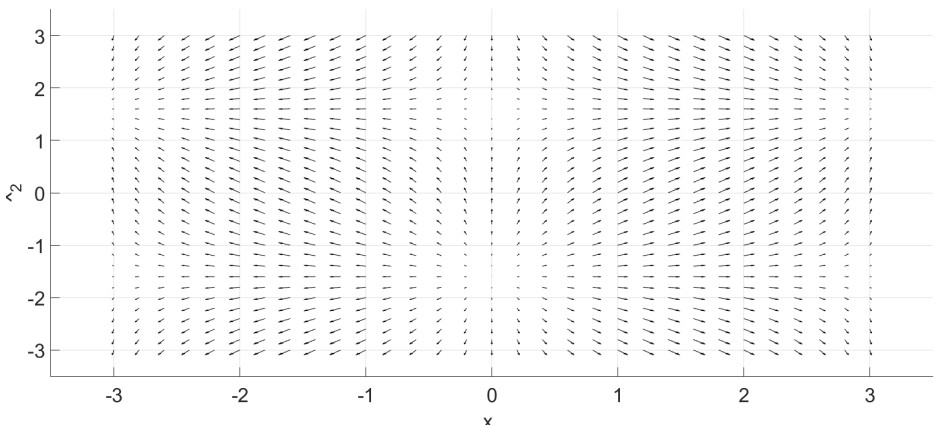

**Figure 4.** Phase portrait view $x_1 - x_2$ for the system (4).

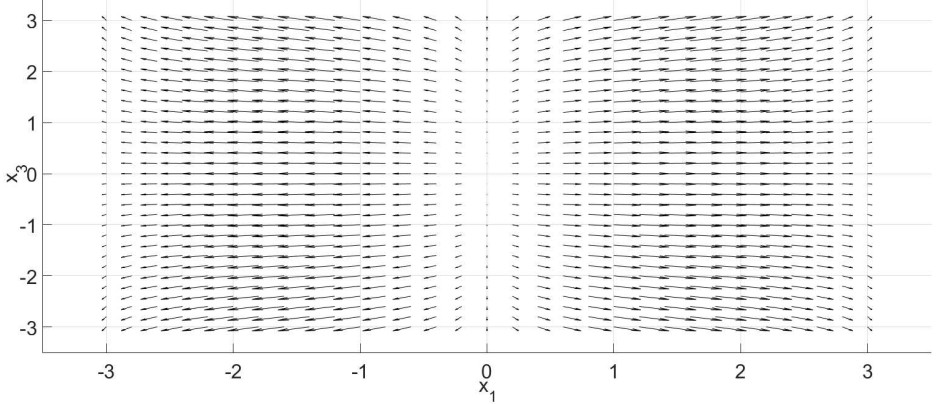

**Figure 5.** Phase portrait view $x_1 - x_3$ for the system (4).

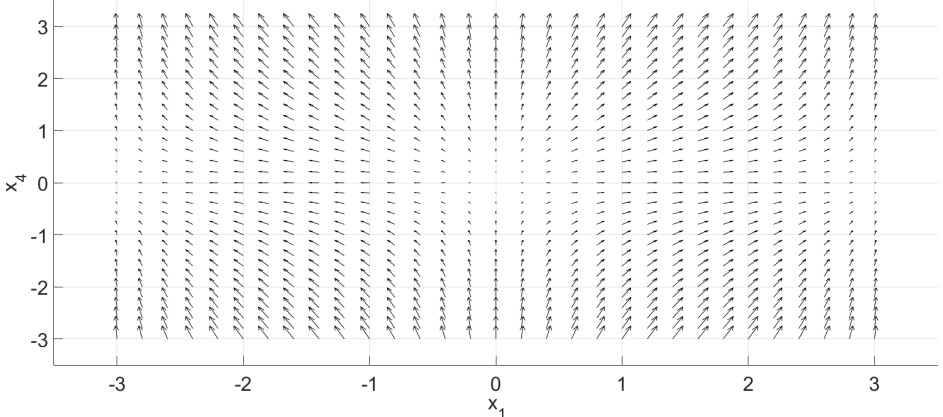

**Figure 6.** Phase portrait view $x_1 - x_4$ for the system (4).

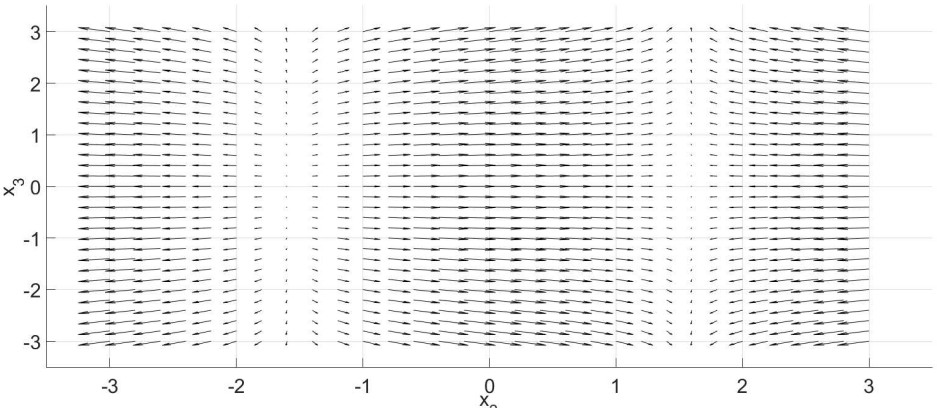

**Figure 7.** Phase portrait view $x_2 - x_3$ for the system (4).

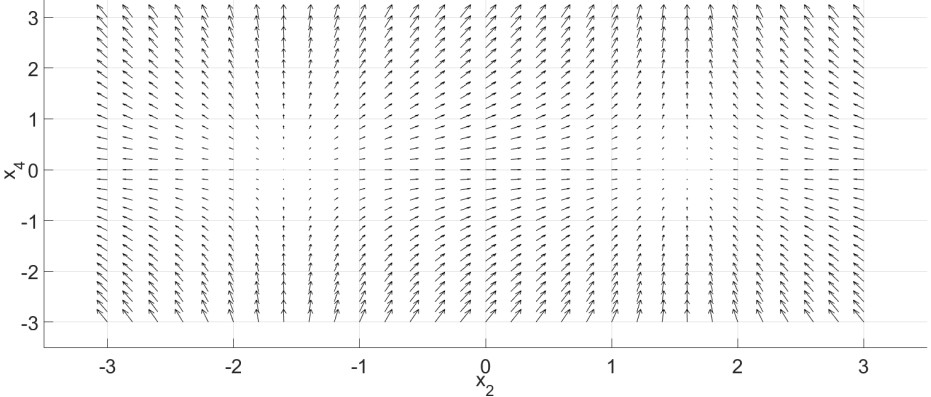

**Figure 8.** Phase portrait view $x_2 - x_4$ for the system (4).

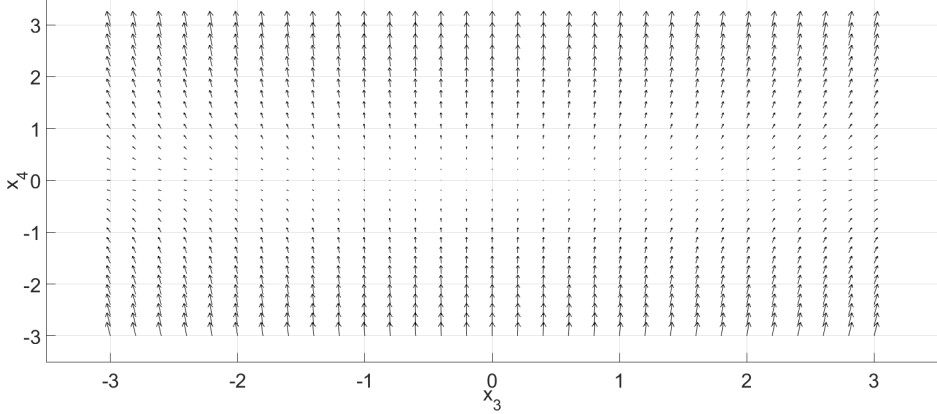

**Figure 9.** Phase portrait view $x_3 - x_4$ for the system (4).

## 3. The n-Dimensional Phase Portraits by Coordinates' Transformations

A coordinate transformation can be performed in order to reduce $v$; however, such strategy can imply a major loss of information if performed arbitrarily. A class of transformations aimed to capture the essentials of a dynamical system by truncating the original system in an appropriate basis are the model order reduction transformations [18]. Although this order reduction is appropriate only in very specific situations, it is worth to mention that with this approach one can show that certain dynamics can be safely neglected.

### 3.1. Foundation

To illustrate this idea, consider the system:

$$\frac{dx}{dt} = f(x) \tag{5}$$

and a transformation $T$ such that $\bar{x} = Tx$, $\bar{x} = [\hat{x}\,\tilde{x}]^T$, $\hat{x} \in \mathbb{R}^k$, $T = [W\,T_2]^T$, $T^{-1} = [V\,T_1]$, such that $VW^T$ is a projection along the kernel of $W$ spanned by $V$:

$$\bar{x} = \begin{bmatrix} \hat{x} \\ \tilde{x} \end{bmatrix} = \begin{bmatrix} W \\ T_2 \end{bmatrix} \begin{bmatrix} V & T_1 \end{bmatrix} \begin{bmatrix} \hat{x} \\ \tilde{x} \end{bmatrix} = \begin{bmatrix} W\left(V\hat{x} + T_1\tilde{x}\right) \\ T_2\left(V\hat{x} + T_1\tilde{x}\right) \end{bmatrix} \tag{6}$$

If the new state vector is replaced in (5) one has:

$$\frac{d\hat{x}}{dt} = W\widehat{f}\left(V\hat{x} + T_1\tilde{x}\right) \tag{7}$$

If the term $T_1\tilde{x}$ is small enough, it can be neglected and a reduction of order that preserves the most important dynamics is obtained:

$$\frac{d\hat{x}}{dt} = W\widehat{f}\left(V\hat{x}\right) \tag{8}$$

The construction of such projection is not a trivial task for a nonlinear system, and it depends on its particular characteristics. Mostly, linearization methods are used to ensure a local proper reduction [19–21]. For linear systems, criteria about the eigenvalues [22], passivity [23,24], Padé approximations, Arnoldi/Krylov methods [25–27], among others are regularly used.

In this paper, linearization/multi-linearization and largest-eigenvalues view reduction is proposed because of its simplicity, enough to plot phase portraits with univocally determined representative views instead of using the original $v$-views phase portraits as in the previous section. That is, the other reduction methods are not a trivial task and they are not always possible.

Consider the linearization of (5) in some initial condition:

$$\frac{dx}{dt} = Ax \tag{9}$$

The state space transformation $(\bar{x} = Tx)$ may be selected as $AT = T\Lambda$ where $T$ is an invertible matrix and $T^{-1}AT$ is a diagonal matrix consisting of the eigenvalues of $A$ from which dominant eigenvalues can be selected to truncate the rest. This process is called modal truncation [28]. If such a transformation can not be achieved, it is possible to use another truncation method as singular decomposition [29], controllability and observability Gramians [18], among others [28]. Alternatively, one can try to select the dominant eigenvalues of $A$ as those of maximum absolute value.

### 3.2. Procedure

From the above, an order reduction is possible from the dominant eigenvalues.

**Procedure 2.** *Follow the next steps:*

1. *Construct the state space representation of the system.*
2. *Define regular initial conditions values for each state.*
3. *Linearize if necessary, in every combination of regular initial conditions; alternatively linearize only in an equilibrium point of interest.*
4. *Try to found matrices such that $AT = T\Lambda$; if not possible use another method for order reduction.*
5. *Determine the dominant states and construct a phase portrait for each combination of them.*
6. *If necessary, redefine the initial conditions values and repeat from step 3; that is, get enough sharpness.*
7. *Analyze each phase portrait separately.*

### 3.3. Examples

In power electronics, the power converters are widely used, and new configurations are frequently reported. In [30] is shown that a cascaded connection of basic power converters (buck and boost), increases by two the order of the dynamical model for each power converter added. For example, for a three cascaded buck converter the dimension grows to $n = 6$; considering all of the control inputs as the 50% of the duty cycle and for common parameters in resistors and inductors with a 100 V power source one can obtain the linearized system (by feedback linearization) on $x = 0$:

$$\frac{d\bar{x}}{dt} = A\bar{x} \tag{10}$$

where:

$$A = \begin{bmatrix} 0 & 0 & 0 & -10 & 0 & 0 \\ 0 & 0 & 0 & 5 & -5 & 0 \\ 0 & 0 & 0 & 0 & 5 & -50 \\ 100 & -100 & 0 & 0 & 0 & 0 \\ 0 & 100 & -100 & 0 & 0 & 0 \\ 0 & 0 & 100 & 0 & 0 & -100 \end{bmatrix} \tag{11}$$

In such case, it is of interest to show the largest derivatives of currents and voltages ($x_1$–$x_3$ and $x_4$ to $x_6$ respectively) by phase portraits. A Jordan decomposition $V^{-1}AV = J$ is possible with:

$$V = \begin{bmatrix} -0.0 & 2.9+4.4i & -0.01 & -0.4-1.9i & 2.9-4.4i & -0.4+1.9i \\ -0.04+0.06i & -2.4-3.0i & -0.04-0.06i & -1.5i & -2.4+3.0i & 1.5i \\ 0.5+0.4i & 0.9-0.4i & 0.5-0.4i & 0.9-0.1i & 0.9+0.4i & 0.9+0.1i \\ -0.1+0.04i & -18.1+12.7i & -0.1-0.04i & 3.2-1.7i & -18.1-12.7i & 3.2+1.7i \\ 0.1+1.1i & 6.3-8.1i & 0.1-1.1i & 8.4-3.1i & 6.3+8.1i & 8.4+3.1i \\ 1.0 & 1.0 & 1.0 & 1.0 & 1.0 & 1.0 \end{bmatrix}$$

$$J = \begin{bmatrix} -44.4+49.6i & 0.0 & 0.0 & 0.0 & 0.0 & 0.0 \\ 0.00 & -0.8228-41.52i & 0.0 & 0.0 & 0.0 & 0.0 \\ 0.0 & 0.0 & -44.4-49.69i & 0.0 & 0.0 & 0.0 \\ 0.0 & 0.0 & 0.0 & -4.7-17.4i & 0.0 & 0.0 \\ 0.0 & 0.0 & 0.0 & 0.0 & -0.82+41.5i & 0.0 \\ 0.0 & 0.0 & 0.0 & 0.0 & 0.0 & -4.7+17.4i \end{bmatrix}$$

and from the absolute value, the dominant eigenvalues indicate that the dominant states are $x_1, x_3$, the less dominants are $x_2, x_5$, and the lesser are $x_4, x_6$. In Figures 10–15 are shown the views for all the combinations of the dominant states for completeness.

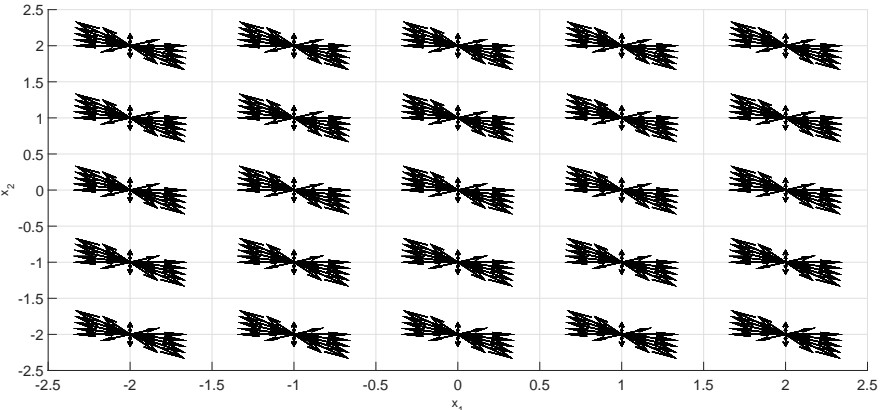

**Figure 10.** Phase portrait, view $x_1$–$x_2$ for the system (10).

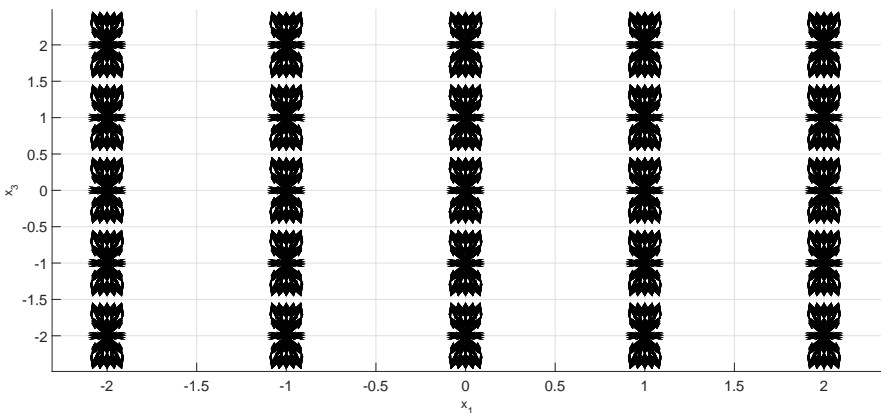

**Figure 11.** Phase portrait, view $x_1$–$x_3$ for the system (10).

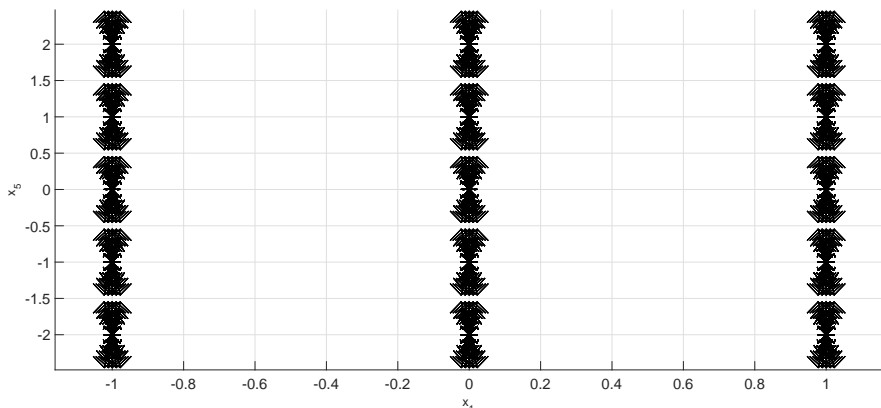

**Figure 12.** Phase portrait, view $x_1$–$x_5$ for the system (10).

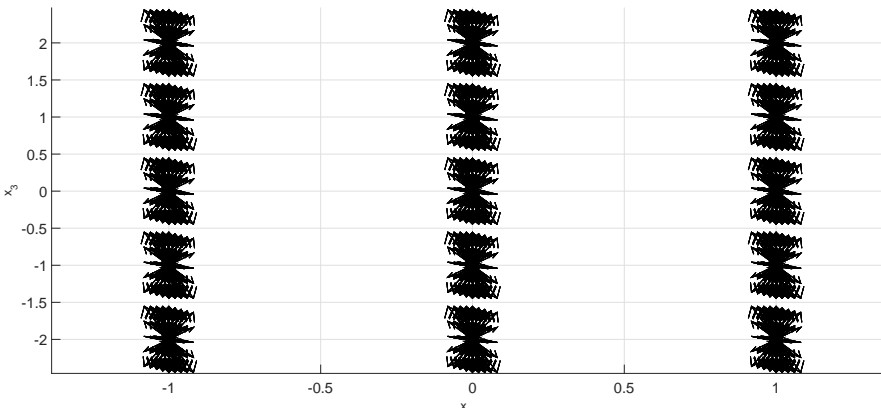

**Figure 13.** Phase portrait, view $x_2$–$x_3$ for the system (10).

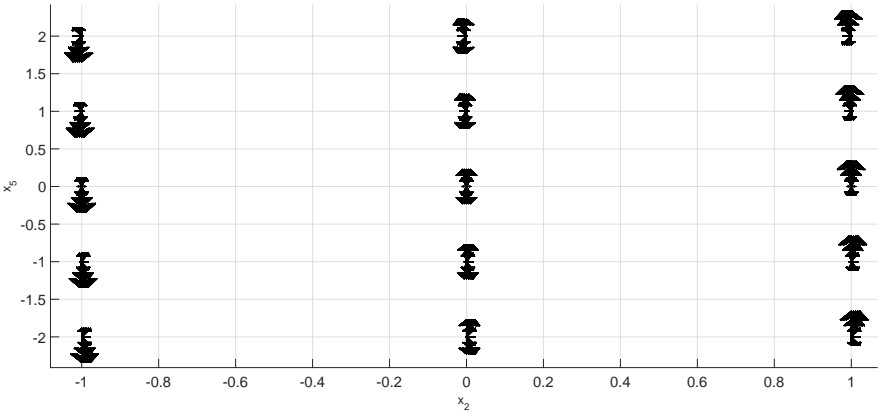

**Figure 14.** Phase portrait, view $x_2$–$x_5$ for the system (10).

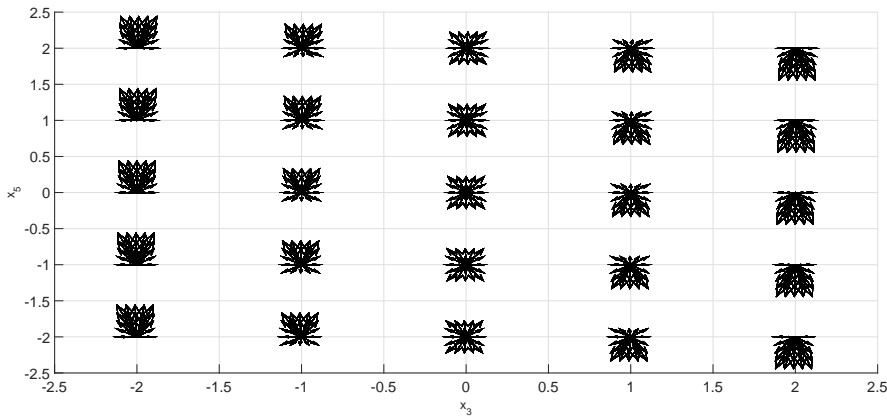

**Figure 15.** Phase portrait, view $x_3$–$x_5$ for the system (10).

## 4. The State-by-State n-Dimensional Phase Portraits

Another approach to represent the behavior of the system dynamics for a set of initial conditions, is looking for the highest magnitudes and signs of the derivatives at each initial condition.

### 4.1. Foundation

In this approach, it is relevant only to look for the magnitude and sign for each state separately, with respect to a set of initial conditions. That is, the *i*- state and its derivatives can be of the same sign, which indicates that the derivative is crescent with respect to the absolute value of the state (indicating possibly instability), or of opposite sign, which indicates that the derivative is decrescent with respect to the absolute value of the state (indicating possibly stability). A manner to accomplish the above is plotting the pairs $[x_i(0), x_i(0)f_i(x(0))]$ where $f_i(x(0))$ is the *i*-th function of the system; this process is repeated for each state variable.

### 4.2. Procedure

From the above, analyzing separately each state, one can plot a phase portrait as described in the following.

**Procedure 3.** *Follow the next steps:*

1.  *Construct the state space representation of the system.*
2.  *Define regular initial conditions values for each state.*
3.  *Calculate $f_i(x(0))$ for each i-th state, and for the set of initial conditions.*
4.  *Plot $[x_i(0), x_i(0)f_i(x(0))]$*
5.  *Repeat Step 3 for each state, a total of n.*
6.  *If necessary, redefine the initial conditions' values and repeat from step 5; that is, get enough sharpness.*
7.  *Analyze each phase portrait separately.*

### 4.3. Examples

Consider again the system (2). Following the previous procedure, a set of 40 values from $-20$ to 20 in increments of one is proposed for each state variable. Fixing $x_1(0) = -20$, one gets a set of values which can be plotted with a vertical line from the point $[x_1(0), 0]$ to the point $[v, 0]$ where $v$ is the velocity (quiver function in Matlab) such that the bigger vertical line overlaps all others (vectors). Repeating the above for the rest of the initial conditions values one gets a plot as the show in Figure 16a in black color (upper plot), and repeating the entire process for the second and third state variables one can obtain the blue (middle) and red (bottom) plots respectively. In this figure, the height of the lines is proportional to the increasing/decreasing rate while the up vertical orientations means an

increasing rate and down vertical orientation means a decreasing rate; that is, a line from zero to a positive vertical value indicates an increasing-value dynamics. It is for the above that for the set of initial conditions one can conclude that all trajectories converge to the equilibrium.

Another approach, shown in Figure 16b is to plot $[x_i(0), x_i(0)f_i(x(0))]$ and similarly to the previous plot Figure 16a, any point under the horizontal axis means a decreasing rate with proportional value to the distance from the axis; the points are joined with a spline.

The state by state phase portrait for (3) is presented in Figure 17. In this case, positive and negative increasing rates reveal the unpredictable behavior typically found in chaotic systems.

Figure 18 shows the state by state phase portrait for the system (4). Although positive and negative increasing rates are present, the behavior is oscillatory instead of chaotic/stable. Note that one can define classes of phase portraits as in the case of 2 dimensional (saddle, bifurcation, etc.).

The instructions used for generating a state by state phase portrait in Matlab are presented in Appendix B.

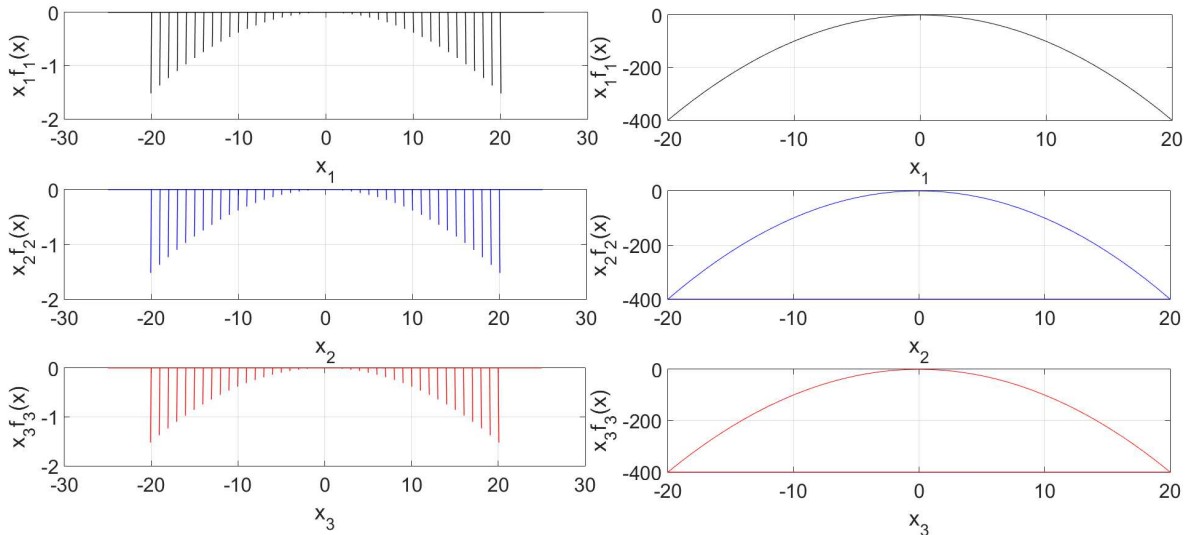

**Figure 16.** State by state phase portrait by (**a**) vectors (left plots) and (**b**) spline (right plots), for system (2).

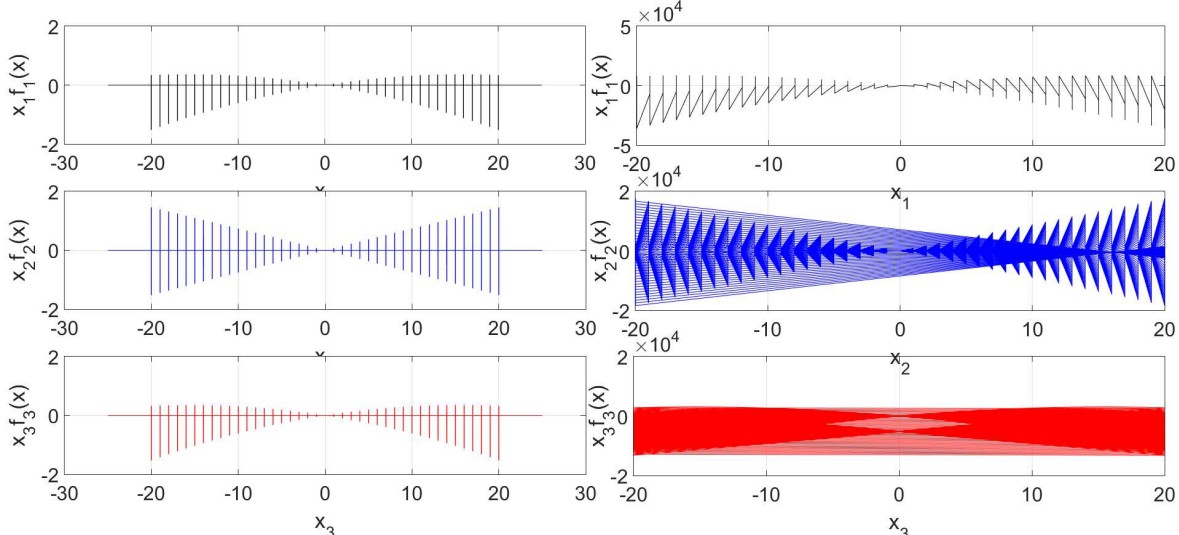

**Figure 17.** State by state phase portrait by (**a**) vectors (left plots) and (**b**) spline (right plots), for system (3).

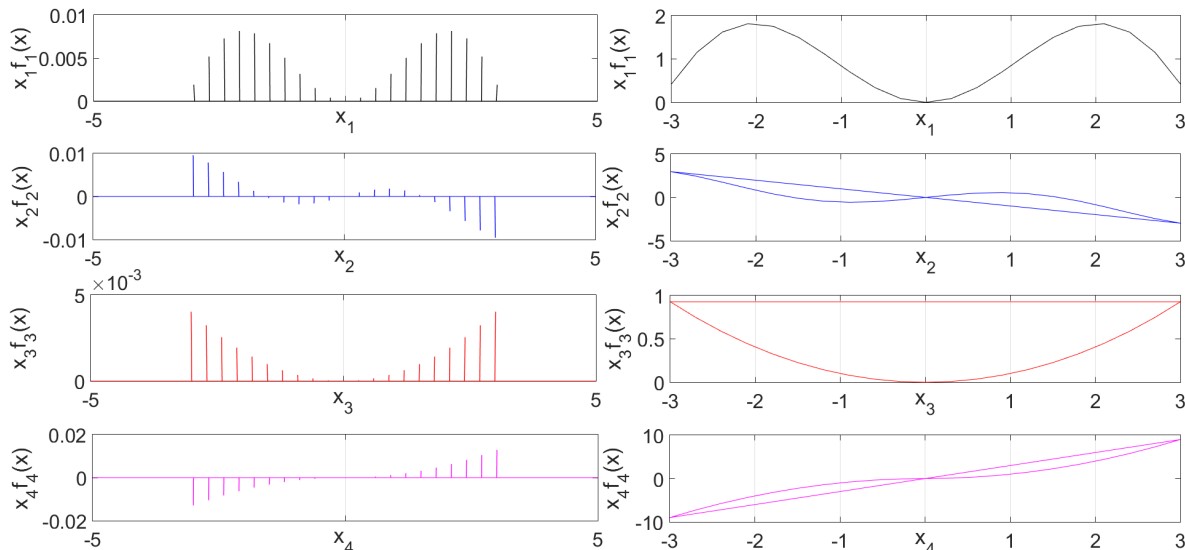

**Figure 18.** State by state phase portrait by (**a**) vectors (left plots) and (**b**) spline (right plots), for system (4).

## 5. Conclusions

In this paper, a study for the construction of n-dimensional phase portraits is performed. Several approaches are presented to formalize and construct phase portraits for systems of higher order. It is demonstrated that the main approach of this paper, the n-dimensional phase portraits by state combinations, is enough to fully illustrate the dynamics in the sense of phase portraits without loss of information, and alternatives to simplify such phase portraits are presented.

**Author Contributions:** Conceptualization, M.-A.R.-L. and F.P.; methodology, M.-A.R.-L.; software, M.-A.R.-L.; validation, M.-A.R.-L., F.-J.P.-P., J.-C.N.-P. and Y.S.-I.; formal analysis, M.-A.R.-L.; investigation, M.-A.R.-L.; resources, M.-A.R.-L., J.-C.N.-P. and Y.S.-I.; writing–original draft preparation, M.-A.R.-L.; writing–review and editing, M.-A.R.-L., F.-J.P.-P., J.-C.N.-P. and Y.S.-I.; funding acquisition, M.-A.R.-L., F.-J.P.-P., J.-C.N.-P. and Y.S.-I.

**Funding:** The authors wish to thank the IPN for its support provided through the project SIP-20190055. In addition, the authors would like to express their gratitude to the COFAA for its financial support and to the CONACYT for Cátedra ID 4155.

**Conflicts of Interest:** The authors declare no conflict of interest for this paper.

## Appendix A. Matlab Code to Generate a n-Dimensional Phase Portrait by State Combinations

```
%%%%%%%%%%%%%%%%% Quiver scale %%%%%%%%%%%%%%%%%%%%%%%%%%%%%%%%%
scale=15;
%%%%%%%%%%%%%%%%% Generate grid %%%%%%%%%%%%%%%%%%%%%%%%%%%%%%%%%
x=permn(-3:3/15:3,4); %(c) Jos van der Geest library from Matlab
%%%%%%%%%%%%%%%%%% THIS IS THE DYNAMICS %%%%%%%%%%%%%%%%%%%%%%%%%
f=[sin(x(:,1)),cos(x(:,2)),tan(x(:,3)/10),abs(x(:,4))];
%%%%%%%%%%%%%%%%%%%%%%%%%%%%%%%%%%%%%%%%%%%%%%%%%%%%%%%%%%%%%%%%%
close all

figure('Name','Phase Portrait x_1 x_2','NumberTitle','off')
hold on
quiver(x(:,1),x(:,2),f(:,1),f(:,2),scale,'k')
xlabel('x_1','FontSize',18)
ylabel('x_2','FontSize',18)
grid on
```

```
figure('Name','Phase Portrait x_1 x_3','NumberTitle','off')
hold on
scale=30;
quiver(x(:,1),x(:,3),f(:,1),f(:,3),scale,'k')
xlabel('x_1','FontSize',18)
ylabel('x_3','FontSize',18)
grid on

figure('Name','Phase Portrait x_1 x_4','NumberTitle','off')
hold on
scale=30;
quiver(x(:,1),x(:,4),f(:,1),f(:,4),scale,'k')
xlabel('x_1','FontSize',18)
ylabel('x_4','FontSize',18)
grid on

figure('Name','Phase Portrait x_2 x_3','NumberTitle','off')
hold on
scale=30;
quiver(x(:,2),x(:,3),f(:,2),f(:,3),scale,'k')
xlabel('x_2','FontSize',18)
ylabel('x_3','FontSize',18)
grid on

figure('Name','Phase Portrait x_2 x_4','NumberTitle','off')
hold on
scale=30;
quiver(x(:,2),x(:,4),f(:,2),f(:,4),scale,'k')
xlabel('x_2','FontSize',18)
ylabel('x_4','FontSize',18)
grid on

figure('Name','Phase Portrait x_3 x_4','NumberTitle','off')
hold on
scale=30;
quiver(x(:,3),x(:,4),f(:,3),f(:,4),scale,'k')
xlabel('x_3','FontSize',18)
ylabel('x_4','FontSize',18)
grid on

shg
```

## Appendix B. Matlab Code to Generate a State by State n-Dimensional Phase Portrait

```
%%%%%%%%%%%%%%%%% Quiver scale %%%%%%%%%%%%%%%%%%%%%%%%%%%%%%%%%%
 scale=1;
%%%%%%%%%%%%%%%%% Generate grid %%%%%%%%%%%%%%%%%%%%%%%%%%%%%%%%%%
x=permn(-3:3/10:3,4); %(c) Jos van der Geest library from Matlab
%%%%%%%%%%%%%%%%% THIS IS THE DYNAMICS %%%%%%%%%%%%%%%%%%%%%%%%%%%%%%
 f=[sin(x(:,1)),cos(x(:,2)),tan(x(:,3)/10),abs(x(:,4))];
```

```
%%%%%%%%%%%%%%%%%%%%%%%%%%%%%%%%%%%%%%%%%%%%%%%%%%%%%%%%%%%%%%%%%%%%%%%%%%%%%%%
s=size(x(:,1));
y=zeros(s(:,1),1);
z=1:1:s(:,1);
z=z';
close all

subplot(4,1,1)
quiver( x(:,1) , y , x(:,1) , x(:,1).*f(:,1) ,scale ,'k','ShowArrowHead',
'off','MaxHeadSize',0.05,'AlignVertexCenters','on','LineWidth',1)
hold on;
line([-5 5],[0 0],'Color','k')

subplot(4,1,2)
quiver( x(:,2) , y , x(:,2) , x(:,2).*f(:,2) ,scale ,'b','ShowArrowHead',
'off','MaxHeadSize',0.05,'AlignVertexCenters','on','LineWidth',1)
hold on;
line([-5 5],[0 0],'Color','b')

subplot(4,1,3)
quiver( x(:,3) , y , x(:,3) , x(:,3).*f(:,3) ,scale ,'r','ShowArrowHead',
'off','MaxHeadSize',0.05,'AlignVertexCenters','on','LineWidth',1)
hold on;
line([-5 5],[0 0],'Color','r')

subplot(4,1,4)
quiver( x(:,4) , y , x(:,4) , x(:,4).*f(:,4) ,scale ,'m','ShowArrowHead',
'off','MaxHeadSize',0.05,'AlignVertexCenters','on','LineWidth',1)
hold on;
line([-5 5],[0 0],'Color','m')

figure
subplot(4,1,1)
plot(x(:,1),x(:,1).*f(:,1),'k')
subplot(4,1,2)
plot(x(:,2),x(:,2).*f(:,2),'b')
subplot(4,1,3)
plot(x(:,3),x(:,3).*f(:,3),'r')
subplot(4,1,4)
plot(x(:,4),x(:,4).*f(:,4),'m')
```

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
