# Peer review of "On the n-Dimensional Phase Portraits"

_applsci, doi:10.3390/app9050872_

Reviewer 1 Report

Review of the article applsci-451389-v1 “On the n-dimensional phase portraits” M.A. Rodríguez-Licea, F.J. Perez-Pinal, J.C. Nuñez Pérez and Y. Sandoval-Ibarra.

The paper can be published in the Applied Sciences journal with a few editorial corrections.

- Section 2.1, pag. 3, line 108. “this is, every” should read “that is, every

- Section 3.1, pag. 9, line 196. “this is, every” should read “that is, every

- Section 3.3, pag. 10, line 220. “For example for a” should read “For example, for a

- Ref. [16], pag. 17, line 314. “briefs” should read “Briefs

Author Response

Honorable reviewer
The authors sincerely appreciate all your suggestions and comments. We consider that the changes requested are pertinent and have been made.
Best regards

Reviewer 2 Report

This is an interesting paper. The authors improved the previous version according to my suggestions.

Concluding: I recommend it for publication as it stands.

Author Response

Honorable reviewer
The authors sincerely appreciate all your valuable suggestions and comments.
Best regards

This manuscript is a resubmission of an earlier submission. The following is a list of the peer review reports and author responses from that submission.

Round  1

Reviewer 1 Report

 The reviewer does not see any new idea or method in the field of dynamical systems; neither does there appear a new intriguing example. 

By the way, Section 2.1 which covers two manuscript pages, could be shortened to one sentence: The number of different 2D planes of an n-dimensional phase space is equal to the number of combinations to select different subsets (xi,  xk) out of the set

(x1,x2,....xn)., which is n! / 2! / (n-2)!.

Author Response

Response to the suggestions and comments on Manuscript ID applsci-405974 entitled “On the n-dimensional Phase Portraits”

The authors thank the honorable Editor-in-Chief (EIC) and Associate Editor (AE) for giving an opportunity to incorporate the valuable suggestions given by reviewers, thereby improving the quality of the paper. The suggestions given by the reviewers are incorporated in the revised manuscript. The authors hope this revision will make our manuscript to meet the requirements of the journal. Changes due to the suggestions and comments of the reviewers are marked in yellow on the new version of the paper and in the following we answer to all the comments of the reviewers; the reviewers’ comments are numbered and indicated with bold and blue fonts as originally were sent.

Reply to Honorable Reviewer 1.

The authors thank the Honorable Reviewer for his excellent review of the paper and in-depth suggestions made to improve the quality of the paper. Followings are the responses to the suggestions of the Honorable Reviewer. The valuable suggestions and corrections are incorporated cautiously, and the paper is made clearer while revising the manuscript.

1.- The reviewer does not see any new idea or method in the field of dynamical systems; neither does there appear a new intriguing example.

Honorable Reviewer, thank you for your comment. In the new version of the article, we have emphasized the main contribution and the examples since, based on the comments of the reviewers, we believe that it has not been clear in the previous version. It should be noted that the authors have not found concrete efforts in the formalization of the phase portraits for higher orders, nor has even a clear method been found for dimensions greater than three. The authors would kindly thank the honorable reviewer for indicating references where the method used in this article is formalized.

2.- By the way, Section 2.1 which covers two manuscript pages, could be shortened to one sentence: The number of different 2D planes of an n-dimensional phase space is equal to the number of combinations to select different subsets (xi,  xk) out of the set.

In the new version of the article, we have emphasized the need to establish visualization concepts from the perspective and analysis that a human observer is capable of performing. That is, it is not enough to remember the number of combinations if not formalize the ability of an analyst to observe phenomena of multi-dimensional spaces. It is for the above, and based on the comments of the reviewers, that a thorough revision of the wording was made in the aforementioned section.

Reply to Honorable Reviewer 2.

The authors thank the Honorable Reviewer for his excellent review of the paper and in-depth suggestions made to improve the quality of the paper. Followings are the responses to the suggestions of the Honorable Reviewer. The valuable suggestions and corrections are incorporated cautiously, and the paper is made clearer while revising the manuscript.

1.- The problem considered in the paper is really interesting. However, explanation of the method is rather poor and can be understood only for a narrow group of specialists. I propose to elucidate the considerations.

Honorable Reviewer, thank you for your comment. In the new version of the article, we have made an effort to clearly describe the contribution and the examples. According to the comments of the reviewers, we believe that we have not been clear in the previous version such that we tried to provide a more didactic manuscript that can be used by many students and professionals.

Reply to Honorable Reviewer 3.

The authors thank the Honorable Reviewer for his excellent review of the paper and in-depth suggestions made to improve the quality of the paper. Followings are the responses to the suggestions of the Honorable Reviewer. The valuable suggestions and corrections are incorporated cautiously, and the paper is made clearer while revising the manuscript. We sorry about the missing line numbering in the first version of the paper, the MDPI template provided does not include it.

1.- The contribution made to literature should be highlighted better in the revised manuscript. It is not clear what the proposed approach can do that cannot be done by the methods available in literature and how this was achieved by the present approach.

Honorable Reviewer, thank you for your comment. In the new version of the article, we have made an effort to clearly describe the contribution and the examples, since, according to the comments of the reviewers, we believe that they have not been clear in the previous version.

2.- Following the previous comment, the proposed approach should be described trying to point out step by step differences from available methods.

Honorable Reviewer, thank you for your comment. In the new version of the paper, we have compared the procedure with that of classic 2D phase portraits and also described with detail the construction of the first example and how the procedure is a generalization of the classic phase portraits.

3.- Provide some algebraic passages to clarify derivation of Eq. (16).

Honorable Reviewer, thank you for your comment. A new expression was added before, in order to clarify the equation (16).

4.- Sections 4.1, 4.2 and 4.3, pp. 11, 12 and 13. “fi(x(0)” should read “fi(x(0))”.

Honorable Reviewer, thank you for spot this. This is corrected in the new version of the paper.

5.- Since vertical axes of left plots in Figure 16 represent velocity values, they should have a different caption from the vertical of right plots. The same holds for Figure 17 and Figure 18.

Honorable Reviewer, thank you for your comment; we agree that seems incorrect. Both (left and right) plots are velocities but plotted by vertical vectors/lines from the horizontal axis and by joining the points (like a spline). In the new version of the paper we have clarified such ideas.

6.- References should be listed more precisely. Please implement the following changes in the revised article.

- Refs. [5,25]. Use capital initial letters only for the first title word of journal articles and conference proceedings;

- Ref. [1,2,4,16,22,23,26,27]. Use capital initial letters for each word of book titles;

- Ref. [17,20,28]. Use capital initial letters for each title word of refereed journals.

Honorable Reviewer, thank you for your comments. These changes have been made in the new version of the article.

7.- Style and grammar must be improved. The authors are kindly requested to check the consistency of the proposed changes with the original content of their article.

- Abstract, pag. 1, line 7. “the overall dynamic” should read “the overall dynamics”

- Introduction, pag. 1, line 6. “in representing and analyzing” should read “to represent and analyze”

- Introduction, pag. 1, line 8. “in better conditions” should read “in the best conditions”

- Introduction, pag. 1, line 12. “tools is based” should read “tool is based”

- Introduction, pag. 2, line 3. “dynamic with” should read “dynamics with”

- Introduction, pag. 2, lines 3 & 15. “In this paper” should read “In that study”

- Introduction, pag. 2, line 6. The sentence “a set of arrows with proportion and direction to the derivative are originated” should be rewritten more clearly

- Introduction, pag. 2, line 14. What does “as a min tool” mean? (order reduction)

- Introduction, pag. 2, line 15. “interesting proposal was reported” should read “interesting approach was proposed”

- Introduction, pag. 2, lines 22-23. “This proposal was focused in piecewise... represented as” should read “The study regarded piecewise... represented them as”

- Introduction, pag. 2, line 24. “Limitation of the” should read “Limiting the capacity of the”

- Introduction, pag. 2, line 27. “of this paper was” should read “of that study was”

- Introduction, pag. 2, line 29. “was reported” should read “was analyzed”

- Introduction, pag. 2, line 31. “Unfortunately... were missed” should read “However... were not documented”

- Introduction, pag. 2, lines 32 & 33. “previous paragraph... dynamic” should read “previous discussion... dynamics”

- Introduction, pag. 2, lines 34 & 35. “As well as a systematical proposal to represent... it has not” should read “Furthermore, a systematical approach for representing... has not”

- Introduction, pag. 2, line 36. “combinations” should read “using combinations”

- Introduction, pag. 2, line 40. “as follow” should read “as follows”

- Introduction, pag. 2, lines 41 & 42. “Section II... On the other hand, Section III... Section IV” should read “Section 2... Section 3... Section 4”

- Section 2, pag. 2, line 45. “system dynamic with” should read “system dynamics with”

- Section 2, pag. 2, line 46. “However, is” should read “However, it is”

- Section 2, pag. 2, line 48. The sentence “and a set of arrows with proportion and direction to the derivative” should be rewritten more clearly

- Section 2, pag. 3, line 2. “and the writer” should read “and one”

- Section 2.3, pag. 5, line 11. “portrait of this paper, allows” should read “portrait presented in this study allows”

- Section 2.3, pag. 5, line 15. “In Annex 5 is shown” should read “Appendix 1 presents”

- Section 3, pag. 6, line 2. “As used previously, instead of (6)” should read “Besides Eq. (6)”

- Section 3.1, pag. 9, line 2. “because its simplicity” should read “because of its simplicity”

- Section 4.1, pag. 11, line 8. “es the” should read “is the”

 - Caption of Figure 16, pag. 13. “(left plots) b) X-Y plotting” should read “(left plots) and b) X-Y plotting (right plots)”

- Caption of Figure 17, pag. 13. “velocity b) X-Y plotting” should read “velocity (left plots) and b) X-Y plotting (right plots)”

- Section 4.3, pag. 13, line 8. “rates shown an” should read “rates reveal the”

- Section 4.3, pag. 13, line 9. “In Figure 18 is shown” should read “Figure 18 shows”

- Section 4.3, pag. 12, line 12. “A sketch of code, to generate... Annex 2” should read “The instructions used for generating... appendix 2”

- Caption of Figure 18, pag. 14. “velocity b) X-Y plotting” should read “velocity (left plots) and b) X-Y plotting (right plots)”

Honorable Reviewer, thank you for your comments. All these comments have been addressed in the new version of the article, thank you sincerely.

Reviewer 2 Report

The problem considered in the paper is really interesting. However explenation of the method is rather poor and can be understood only for a narrow  group of specialists. I propose to elucidate the considerations.

After this I will warmly recommend the paper for publication in Applied Sciences.

Author Response

(The authors gave the same response as above.)

Reviewer 3 Report

Review of the article applsci-405974-v1 “On the n-dimensional phase portraits” M.A. Rodríguez-Licea, F.J. Perez-Pinal, J.C. Nuñez Pérez and Y. Sandoval-Ibarra.

The article presented methods to represent phase distribution of high order dynamical systems and proposed a new approach for n-dimension systems.

The subject of the article appears interesting but the article needs major revisions before final publication in the Applied Sciences journal. In particular, the contribution of the study should be highlighted better and the proposed approach should be clarified in the revised manuscript. Style and grammar of the article should be improved.

Some suggestions on how to improve the article are given below. Since lines were not numbered in the original submission, “line 1” will correspond the first text line (excluding equations and figure captions) of each page.

Ø The contribution made to literature should be highlighted better in the revised manuscript. It is not clear what the proposed approach can do that cannot be done by the methods available in literature and how this was achieved by the present approach.

Ø Following the previous comment, the proposed approach should be described trying to point out step by step differences from available methods.

Ø Provide some algebraic passages to clarify derivation of Eq. (16).

Ø Sections 4.1, 4.2 and 4.3, pp. 11, 12 and 13. “fi(x(0)” should read “fi(x(0))”.

Ø Since vertical axes of left plots in Figure 16 represent velocity values, they should have a different caption from the vertical of right plots. The same holds for Figure 17 and Figure 18.

Ø References should be listed more precisely. Please implement the following changes in the revised article.

- Refs. [5,25]. Use capital initial letters only for the first title word of journal articles and conference proceedings;

- Ref. [1,2,4,16,22,23,26,27]. Use capital initial letters for each word of book titles;

- Ref. [17,20,28]. Use capital initial letters for each title word of refereed journals.

Ø Style and grammar must be improved. The authors are kindly requested to check the consistency of the proposed changes with the original content of their article.

- Abstract, pag. 1, line 7. “the overall dynamic” should read “the overall dynamics”

- Introduction, pag. 1, line 6. “in representing and analyzing” should read “to represent and analyze

- Introduction, pag. 1, line 8. “in better conditions” should read “in the best conditions

- Introduction, pag. 1, line 12. “tools is based” should read “tool is based

- Introduction, pag. 2, line 3. “dynamic with” should read “dynamics with

- Introduction, pag. 2, lines 3 & 15. “In this paper” should read “In that study

- Introduction, pag. 2, line 6. The sentence “a set of arrows with proportion and direction to the derivative are originated” should be rewritten more clearly

- Introduction, pag. 2, line 14. What does “as a min tool” mean?

- Introduction, pag. 2, line 15. “interesting proposal was reported” should read “interesting approach was proposed

- Introduction, pag. 2, lines 22-23. “This proposal was focused in piecewise... represented as” should read “The study regarded piecewise... represented them as

- Introduction, pag. 2, line 24. “Limitation of the” should read “Limiting the capacity of the

- Introduction, pag. 2, line 27. “of this paper was” should read “of that study was

- Introduction, pag. 2, line 29. “was reported” should read “was analyzed

- Introduction, pag. 2, line 31. “Unfortunately... were missed” should read “However... were not documented

- Introduction, pag. 2, lines 32 & 33. “previous paragraph... dynamic” should read “previous discussion... dynamics

- Introduction, pag. 2, lines 34 & 35. “As well as a systematical proposal to represent... it has not” should read “Furthermore, a systematical approach for representing... has not

- Introduction, pag. 2, line 36. “combinations” should read “using combinations

- Introduction, pag. 2, line 40. “as follow” should read “as follows

- Introduction, pag. 2, lines 41 & 42. “Section II... On the other hand, Section III... Section IV” should read “Section 2... Section 3... Section 4

- Section 2, pag. 2, line 45. “system dynamic with” should read “system dynamics with

- Section 2, pag. 2, line 46. “However, is” should read “However, it is

- Section 2, pag. 2, line 48. The sentence “and a set of arrows with proportion and direction to the derivative” should be rewritten more clearly

- Section 2, pag. 3, line 2. “and the writer” should read “and one

- Section 2.3, pag. 5, line 11. “portrait of this paper, allows” should read “portrait presented in this study allows

- Section 2.3, pag. 5, line 15. “In Annex 5 is shown” should read “Appendix 1 presents

- Section 3, pag. 6, line 2. “As used previously, instead of (6)” should read “Besides Eq. (6)

- Section 3.1, pag. 9, line 2. “because its simplicity” should read “because of its simplicity

- Section 4.1, pag. 11, line 8. “es the” should read “is the

- Caption of Figure 16, pag. 13. “(left plots) b) X-Y plotting” should read “(left plots) and b) X-Y plotting (right plots)

- Caption of Figure 17, pag. 13. “velocity b) X-Y plotting” should read “velocity (left plots) and b) X-Y plotting (right plots)

- Section 4.3, pag. 13, line 8. “rates shown an” should read “rates reveal the

- Section 4.3, pag. 13, line 9. “In Figure 18 is shown” should read “Figure 18 shows

- Section 4.3, pag. 12, line 12. “A sketch of code, to generate... Annex 2” should read “The instructions used for generating... appendix 2

- Caption of Figure 18, pag. 14. “velocity b) X-Y plotting” should read “velocity (left plots) and b) X-Y plotting (right plots)

Author Response

Response to the suggestions and comments on Manuscript ID applsci-405974 entitled “On the n-dimensional Phase Portraits”

The authors thank the honorable Editor-in-Chief (EIC) and Associate Editor (AE) for giving an opportunity to incorporate the valuable suggestions given by reviewers, thereby improving the quality of the paper. The suggestions given by the reviewers are incorporated in the revised manuscript. The authors hope this revision will make our manuscript to meet the requirements of the journal. Changes due to the suggestions and comments of the reviewers are marked in yellow on the new version of the paper and in the following we answer to all the comments of the reviewers; the reviewers’ comments are numbered and indicated with bold and blue fonts as originally were sent.

Reply to Honorable Reviewer 1.

The authors thank the Honorable Reviewer for his excellent review of the paper and in-depth suggestions made to improve the quality of the paper. Followings are the responses to the suggestions of the Honorable Reviewer. The valuable suggestions and corrections are incorporated cautiously, and the paper is made clearer while revising the manuscript.

1.- The reviewer does not see any new idea or method in the field of dynamical systems; neither does there appear a new intriguing example.

Honorable Reviewer, thank you for your comment. In the new version of the article, we have emphasized the main contribution and the examples since, based on the comments of the reviewers, we believe that it has not been clear in the previous version. It should be noted that the authors have not found concrete efforts in the formalization of the phase portraits for higher orders, nor has even a clear method been found for dimensions greater than three. The authors would kindly thank the honorable reviewer for indicating references where the method used in this article is formalized.

2.- By the way, Section 2.1 which covers two manuscript pages, could be shortened to one sentence: The number of different 2D planes of an n-dimensional phase space is equal to the number of combinations to select different subsets (xi,  xk) out of the set.

In the new version of the article, we have emphasized the need to establish visualization concepts from the perspective and analysis that a human observer is capable of performing. That is, it is not enough to remember the number of combinations if not formalize the ability of an analyst to observe phenomena of multi-dimensional spaces. It is for the above, and based on the comments of the reviewers, that a thorough revision of the wording was made in the aforementioned section.

Reply to Honorable Reviewer 2.

The authors thank the Honorable Reviewer for his excellent review of the paper and in-depth suggestions made to improve the quality of the paper. Followings are the responses to the suggestions of the Honorable Reviewer. The valuable suggestions and corrections are incorporated cautiously, and the paper is made clearer while revising the manuscript.

1.- The problem considered in the paper is really interesting. However, explanation of the method is rather poor and can be understood only for a narrow group of specialists. I propose to elucidate the considerations.

Honorable Reviewer, thank you for your comment. In the new version of the article, we have made an effort to clearly describe the contribution and the examples. According to the comments of the reviewers, we believe that we have not been clear in the previous version such that we tried to provide a more didactic manuscript that can be used by many students and professionals.

Reply to Honorable Reviewer 3.

The authors thank the Honorable Reviewer for his excellent review of the paper and in-depth suggestions made to improve the quality of the paper. Followings are the responses to the suggestions of the Honorable Reviewer. The valuable suggestions and corrections are incorporated cautiously, and the paper is made clearer while revising the manuscript. We sorry about the missing line numbering in the first version of the paper, the MDPI template provided does not include it.

1.- The contribution made to literature should be highlighted better in the revised manuscript. It is not clear what the proposed approach can do that cannot be done by the methods available in literature and how this was achieved by the present approach.

Honorable Reviewer, thank you for your comment. In the new version of the article, we have made an effort to clearly describe the contribution and the examples, since, according to the comments of the reviewers, we believe that they have not been clear in the previous version.

2.- Following the previous comment, the proposed approach should be described trying to point out step by step differences from available methods.

Honorable Reviewer, thank you for your comment. In the new version of the paper, we have compared the procedure with that of classic 2D phase portraits and also described with detail the construction of the first example and how the procedure is a generalization of the classic phase portraits.

3.- Provide some algebraic passages to clarify derivation of Eq. (16).

Honorable Reviewer, thank you for your comment. A new expression was added before, in order to clarify the equation (16).

4.- Sections 4.1, 4.2 and 4.3, pp. 11, 12 and 13. “fi(x(0)” should read “fi(x(0))”.

Honorable Reviewer, thank you for spot this. This is corrected in the new version of the paper.

5.- Since vertical axes of left plots in Figure 16 represent velocity values, they should have a different caption from the vertical of right plots. The same holds for Figure 17 and Figure 18.

Honorable Reviewer, thank you for your comment; we agree that seems incorrect. Both (left and right) plots are velocities but plotted by vertical vectors/lines from the horizontal axis and by joining the points (like a spline). In the new version of the paper we have clarified such ideas.

6.- References should be listed more precisely. Please implement the following changes in the revised article.

- Refs. [5,25]. Use capital initial letters only for the first title word of journal articles and conference proceedings;

- Ref. [1,2,4,16,22,23,26,27]. Use capital initial letters for each word of book titles;

- Ref. [17,20,28]. Use capital initial letters for each title word of refereed journals.

Honorable Reviewer, thank you for your comments. These changes have been made in the new version of the article.

7.- Style and grammar must be improved. The authors are kindly requested to check the consistency of the proposed changes with the original content of their article.

- Abstract, pag. 1, line 7. “the overall dynamic” should read “the overall dynamics”

- Introduction, pag. 1, line 6. “in representing and analyzing” should read “to represent and analyze”

- Introduction, pag. 1, line 8. “in better conditions” should read “in the best conditions”

- Introduction, pag. 1, line 12. “tools is based” should read “tool is based”

- Introduction, pag. 2, line 3. “dynamic with” should read “dynamics with”

- Introduction, pag. 2, lines 3 & 15. “In this paper” should read “In that study”

- Introduction, pag. 2, line 6. The sentence “a set of arrows with proportion and direction to the derivative are originated” should be rewritten more clearly

- Introduction, pag. 2, line 14. What does “as a min tool” mean? (order reduction)

- Introduction, pag. 2, line 15. “interesting proposal was reported” should read “interesting approach was proposed”

- Introduction, pag. 2, lines 22-23. “This proposal was focused in piecewise... represented as” should read “The study regarded piecewise... represented them as”

- Introduction, pag. 2, line 24. “Limitation of the” should read “Limiting the capacity of the”

- Introduction, pag. 2, line 27. “of this paper was” should read “of that study was”

- Introduction, pag. 2, line 29. “was reported” should read “was analyzed”

- Introduction, pag. 2, line 31. “Unfortunately... were missed” should read “However... were not documented”

- Introduction, pag. 2, lines 32 & 33. “previous paragraph... dynamic” should read “previous discussion... dynamics”

- Introduction, pag. 2, lines 34 & 35. “As well as a systematical proposal to represent... it has not” should read “Furthermore, a systematical approach for representing... has not”

- Introduction, pag. 2, line 36. “combinations” should read “using combinations”

- Introduction, pag. 2, line 40. “as follow” should read “as follows”

- Introduction, pag. 2, lines 41 & 42. “Section II... On the other hand, Section III... Section IV” should read “Section 2... Section 3... Section 4”

- Section 2, pag. 2, line 45. “system dynamic with” should read “system dynamics with”

- Section 2, pag. 2, line 46. “However, is” should read “However, it is”

- Section 2, pag. 2, line 48. The sentence “and a set of arrows with proportion and direction to the derivative” should be rewritten more clearly

- Section 2, pag. 3, line 2. “and the writer” should read “and one”

- Section 2.3, pag. 5, line 11. “portrait of this paper, allows” should read “portrait presented in this study allows”

- Section 2.3, pag. 5, line 15. “In Annex 5 is shown” should read “Appendix 1 presents”

- Section 3, pag. 6, line 2. “As used previously, instead of (6)” should read “Besides Eq. (6)”

- Section 3.1, pag. 9, line 2. “because its simplicity” should read “because of its simplicity”

- Section 4.1, pag. 11, line 8. “es the” should read “is the”

 - Caption of Figure 16, pag. 13. “(left plots) b) X-Y plotting” should read “(left plots) and b) X-Y plotting (right plots)”

- Caption of Figure 17, pag. 13. “velocity b) X-Y plotting” should read “velocity (left plots) and b) X-Y plotting (right plots)”

- Section 4.3, pag. 13, line 8. “rates shown an” should read “rates reveal the”

- Section 4.3, pag. 13, line 9. “In Figure 18 is shown” should read “Figure 18 shows”

- Section 4.3, pag. 12, line 12. “A sketch of code, to generate... Annex 2” should read “The instructions used for generating... appendix 2”

- Caption of Figure 18, pag. 14. “velocity b) X-Y plotting” should read “velocity (left plots) and b) X-Y plotting (right plots)”

Honorable Reviewer, thank you for your comments. All these comments have been addressed in the new version of the article, thank you sincerely.

Round  2

Reviewer 3 Report

Review of the article applsci-405974-v2 “On the n-dimensional phase portraits” M.A. Rodríguez-Licea, F.J. Perez-Pinal, J.C. Nuñez Pérez and Y. Sandoval-Ibarra.

The authors addressed the comments made by this reviewer and improved the quality of their article, which can now be published in the Applied Sciences journal with the following amendments.

Ø References should be listed more precisely. Please implement the following changes in the revised article.

- Ref. [1,2,3,4,5,18,24,25,28,29]. Use capital initial letters for each title word of books and book chapters. For example, Ref. [2], "Modern control systems" should read "Modern Control Systems";

- Ref. [22]. Use capital initial letters for each title word of refereed journals.

Ø Editorial corrections.

- Introduction, pag. 2, line 40. “degraded to a two-dimensional” should read “turned two-dimensional

- Introduction, pag. 2, line 46. “condition, both relies” should read “condition rely

- Introduction, pag. 2, lines 75 & 77. “In summarize… it has not” should read “In summaryyet has not

- Introduction, pag. 2, line 78. “solve these interrogations” should read “address this question

- Introduction, pag. 2, line 79. “using using” should read “using

- Section 2, pag. 3, lines 96-97. “arrows with proportion and direction to the derivative” should read “arrows directed toward derivative and with size proportional to its magnitude

- Section 2, pag. 3, line 100. “this is, one” should read “that is, one

Author Response

Response to the suggestions and comments on Manuscript ID applsci-405974 entitled “On the n-dimensional Phase Portraits”

The authors thank the honorable Editor-in-Chief (EIC) and Associate Editor (AE) for giving an opportunity to incorporate the valuable suggestions given by reviewers, thereby improving the quality of the paper. The suggestions given by the reviewers are incorporated in the revised manuscript. The authors hope this revision will make our manuscript to meet the requirements of the journal. Changes due to the suggestions and comments of the reviewers are marked in yellow on the new version of the paper and in the following we answer to all the comments of the reviewers; the reviewers’ comments are numbered and indicated with bold and blue fonts as originally were sent.

Reply to Honorable Reviewer 3.

The authors sincerely thank the Honorable Reviewer for his excellent, detailed, and in-depth review of the paper made to improve the quality of the paper. The valuable suggestions and corrections are incorporated cautiously.

1.- References should be listed more precisely. Please implement the following changes in the revised article.

- Ref. [1,2,3,4,5,18,24,25,28,29]. Use capital initial letters for each title word of books and book chapters. For example, Ref. [2], "Modern control systems" should read "Modern Control Systems";

- Ref. [22]. Use capital initial letters for each title word of refereed journals.

Honorable Reviewer, thank you for your corrections. We have also verified that the references comply with the MDPI standards.

2.- Editorial corrections.

- Introduction, pag. 2, line 40. “degraded to a two-dimensional” should read “turned two-dimensional”

- Introduction, pag. 2, line 46. “condition, both relies” should read “condition rely”

- Introduction, pag. 2, lines 75 & 77. “In summarize… it has not” should read “In summary… yet has not”

- Introduction, pag. 2, line 78. “solve these interrogations” should read “address this question”

- Introduction, pag. 2, line 79. “using using” should read “using”

- Section 2, pag. 3, lines 96-97. “arrows with proportion and direction to the derivative” should read “arrows directed toward derivative and with size proportional to its magnitude”

- Section 2, pag. 3, line 100. “this is, one” should read “that is, one”

Honorable Reviewer, thank you for your comments. All these comments have been addressed in the new version of the article, thank you sincerely.
